# Batch production of 6-inch uniform monolayer molybdenum disulfide catalyzed by sodium in glass

Pengfei Yang[1,2], Xiaolong Zou[3], Zhepeng Zhang[1,2], Min Hong[1,2], Jianping Shi[1,2], Shulin Chen[2,4], Jiapei Shu[5], Liyun Zhao[1], Shaolong Jiang[1,2], Xiebo Zhou[1,2], Yahuan Huan[1,2], Chunyu Xie[1,2], Peng Gao[2,6,7], Qing Chen [ID] [5], Qing Zhang [ID] [1], Zhongfan Liu[2] & Yanfeng Zhang[1,2]

Monolayer transition metal dichalcogenides (TMDs) have become essential two-dimensional materials for their perspectives in engineering next-generation electronics. For related applications, the controlled growth of large-area uniform monolayer TMDs is crucial, while it remains challenging. Herein, we report the direct synthesis of 6-inch uniform monolayer molybdenum disulfide on the solid soda-lime glass, through a designed face-to-face metal-precursor supply route in a facile chemical vapor deposition process. We find that the highly uniform monolayer film, with the composite domains possessing an edge length larger than 400 µm, can be achieved within a quite short time of 8 min. This highly efficient growth is proven to be facilitated by sodium catalysts that are homogenously distributed in glass, according to our experimental facts and density functional theory calculations. This work provides insights into the batch production of highly uniform TMD films on the functional glass substrate with the advantages of low cost, easily transferrable, and compatible with direct applications.

[1] Department of Materials Science and Engineering, College of Engineering, Peking University, Beijing 100871, China. [2] Center for Nanochemistry (CNC), Academy for Advanced Interdisciplinary Studies, Beijing National Laboratory for Molecular Sciences, College of Chemistry and Molecular Engineering, Peking University, Beijing 100871, China. [3] Tsinghua-Berkeley Shenzhen Institute (TBSI), Tsinghua University, Shenzhen 518055, China. [4] State Key Laboratory of Advanced Welding and Joining, Harbin Institute of Technology, Harbin 150001, China. [5] Key Laboratory for the Physics and Chemistry of Nanodevices, Department of Electronics, Peking University, Beijing 100871, China. [6] Electron Microscopy Laboratory, and International Center for Quantum Materials, School of Physics, Peking University, Beijing 100871, China. [7] Collaborative Innovation Center of Quantum Matter, Beijing 100871, China. Correspondence and requests for materials should be addressed to Y.Z. (email: yanfengzhang@pku.edu.cn)

Recently, two-dimensional transition metal dichalcogenides (TMDs) have opened new perspectives for engineering next-generation electronics and optoelectronics, thanks to their unique physical and chemical properties different from their bulk counterparts[1–5]. In particular, direct band gap semiconductors, such as $MoS_2$ and $WS_2$, exhibit ultrahigh optical responsivity[6], efficient valley polarization[7,8] and strong light–matter coupling[9], making them highly promising materials for constructing electrical/optical and energy-related devices. For such applications, the controllable synthesis of large-area uniform and large-domain monolayer TMDs is highly desired, while still challenging. This challenge is similarly encountered in graphene researches, and many efforts have been made in recent decades[10].

To date, various techniques have been developed to synthesize TMDs, including physical vapor deposition[11], metal organic chemical vapor deposition[12,13], chemical vapor deposition (CVD)[14,15], etc. Among these, the CVD has been recognized as the most promising route for directly synthesizing large-area uniform multi- or monolayer TMD materials[16–25]. Specifically, three strategies have been employed to deliver the metal precursors during the CVD process: the "pre-deposited" route, "point-to-face" supply, and "face-to-face" feeding methods. Taking the synthesis of $MoS_2$ as an example, pre-deposition of Mo layers and $MoO_2$ microcrystals followed by sulfurization processes were developed by Zhan et al.[17] and Wang et al.[18], respectively, for realizing the large-area growth of thin-layer $MoS_2$ films. However, the TMDs usually evolved as polycrystalline layers due to limited migration/diffusion of metal precursors on macroscopic-scale surfaces. Subsequently, a point-to-face metal-precursor supply method was proposed to grow large-domain TMDs with controllable thicknesses. Lee et al. reported the successful synthesis of $MoS_2$ atomic layers on $SiO_2/Si$ by mounting the substrate face-down over the $MoO_3$ powder and sulfur precursors[20]. In addition, the substrate was also spin-coated with graphene-like molecules to promote the layered growth of $MoS_2$. Without using seeds, Najmaei et al. demonstrated the synthesis of triangular domains and continuous $MoS_2$ films by placing $MoO_3$-nanoribbon-covered plate on the silicon substrate to reveal the nucleation and grain boundary formation mechanism[26].

Notably, a point-to-face metal-source supply method has shown great potential for obtaining large-domain TMD monolayers using metal oxides (e.g, $MoO_3$, $MoO_2$) precursors placed upstream of (or below) the substrates. However, achieving large-scale uniform monolayer TMDs (over several centimeters) remains challenging, due to the variable release rates of the metal precursor and its inhomogeneous distribution along the gas-flow direction[27,28]. To guarantee a sufficient and uniform delivery of the metal precursor, a face-to-face deposition method was then proposed. Specifically, Yun et al. reported the direct synthesis of $2 \times 3 \text{ cm}^2$ uniform monolayer $WS_2$ films by arranging $(NH_4)_6H_2W_{12}O_{40} \cdot x(H_2O)$ (AMT)-loaded $Al_2O_3$ plate above Au substrates[29]. However, the feeding rate of the precursors achieved through the two-step solution-processed assembly was still uncontrollable, and the method was too tedious to apply. Chen et al. reported the fast growth of millimeter-size monolayer $MoSe_2$ crystals on molten glass[22]. A piece of Mo foil placed on $SiO_2/Si$ was used to hold the glass substrate. Very recently, the same group achieved the homoepitaxial growth of $MoS_2$ patterns on monolayer $MoS_2$ at a growth temperature of 1050 °C[23]. In the growth process, a piece of curved Mo foil was placed above the molten glass substrate serving as the Mo source. The ionic compounds in glass corroded the Mo foil thus helping with the volatilization of Mo, and resulting in a face-to-face metal-precursor feeding pathway.

Herein, we design a facile face-to-face metal-precursor supply strategy in the conventional CVD growth process, for synthesizing large-scale uniform, monolayer $MoS_2$ by selecting a novel solid glass substrate and using Mo foil and sulfur as precursors. Soda-lime glass is chosen as the substrate as it is cost-effective and scalable, thus suitable for the batch production of monolayer $MoS_2$ films. Meanwhile, the coating of $MoS_2$ on glass endows it with novel optical and catalytic properties. Particularly, the large-scale uniformity, crystallinity, and growth efficiency of $MoS_2$ on soda-lime glass substrate are carefully evaluated via detailed characterizations from atomic to centimeter scales. The ultrafast growth of $MoS_2$ with the aid of trace amount of Na catalysts from the glass substrate is also discussed, according to intensive experimental efforts and density functional theory (DFT) calculations. In addition, by exploiting the hydrophilicity of the glass substrate, we also develop an etching-free transfer process for transferring inch-scale $MoS_2$ films onto target substrates. This work, hereby, provides novel insights into the batch production and transfer of macroscopic uniform TMD films, which will propel their practical applications in various fields.

## Results

**Face-to-face metal-precursor feeding route.** In order to obtain a highly uniform monolayer $MoS_2$ film, we designed a unique face-to-face metal-precursor feeding route. As schematically illustrated in Fig. 1a, a Mo foil was placed above the soda-lime glass substrate in a parallel geometry with a gap of 10 mm. This configuration ensures a homogeneous supply of the Mo precursor through an onsite heating process with the aid of an $O_2$ carrier (Supplementary Fig. 1). The concentration of the S precursor is usually oversaturated during the CVD growth process[30]. For the batch production purpose, commercial soda-lime glass (mainly composed of $SiO_2$, $Na_2O$, and $CaO$) was selected as the growth substrate considering the following factors: first, its low cost and scalability guarantees the cost-effective production of $MoS_2$/glass hybrid materials; second, its hydrophilicity feature facilitates the transfer of $MoS_2$ films to other substrates; third, the $MoS_2$/glass hybrid itself may serve as a prototype material for directly fabricating optical and photoelectric devices due to its relatively high transparency (exceeds 91% at a wavelength $\lambda = 550$ mm). It is worthy of mention that the molten glass has been utilized as an ideal substrate for graphene growth in our previous efforts[31]. In another work, Ju et al. proposed the self-limited monolayer growth mechanism of $MoS_2$ on molten glass for it is capable of trapping overflowing $MoO_3$ precursors[32]. In contrast, the growth temperature in our system is set to 720 °C in order to preserve the original surface morphology of the glass substrate, thus providing higher potential for the batch production and the direct application of the hybrid material.

In the low-pressure (LP) CVD process, an adequate sulfur vapor was conveyed downstream by a flow of Ar (50 sccm) and $O_2$ (6 sccm). The $O_2$ carrier was introduced to oxidize the Mo foil for directly releasing chemically active $MoO_{3-x}$ ($x = 2, 3$). This intermediate precursor presents relatively low sublimation temperature of about 500 °C, right below the melting point of the soda-lime glass. Notably, this temperature is much lower than that needed for the direct release of Mo atoms from Mo foils (~2600 °C). Generally, $MoO_{3-x}$ is regarded as active species for the CVD synthesis of $MoS_2$[19,33]. As shown in Fig. 1b, at relatively short growth time of 8 min, a $14 \times 6 \text{ cm}^2$ uniform, full-covered $MoS_2$ film is synthesized on the glass substrate at 720 °C by a LPCVD strategy (see the Methods section). After this growth process, deformation of the glass substrate is negligible; however, its surface color changes from transparent to light brown, indicating the formation of $MoS_2$.

According to the X-ray photoelectron spectroscopy (XPS) data, the Mo $3d_{5/2}$ (at 229.4 eV) and $3d_{3/2}$ (at 232.6 eV) peaks are in

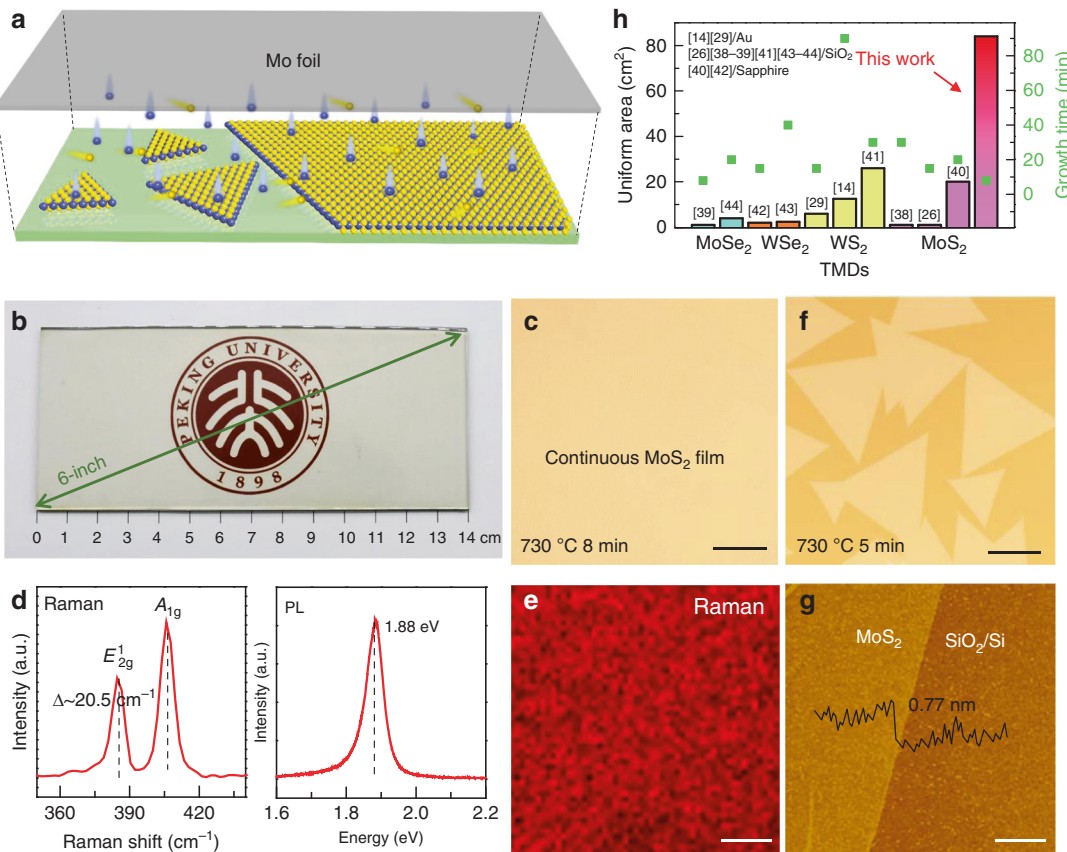

**Fig. 1** LPCVD growth of large-area uniform, monolayer $MoS_2$ on soda-lime glass. **a** Schematic diagram of a face-to-face metal-precursor supply route. **b** Photograph of a 6-inch continuous $MoS_2$ film on soda-lime glass synthesized for ~8 min. **c** Typical OM image of the $MoS_2$ film, scale bar: 0.2 mm. **d** Raman (left panel) and PL (right panel) spectra of the as-grown $MoS_2$, confirming its monolayer feature. **e** Raman mapping on the intensity of $A_{1g}$ peak for the continuous $MoS_2$ film, scale bar: 10 μm. **f** OM image of triangular $MoS_2$ domains on glass grown for ~5 min with the other parameters kept identical. Scale bar: 0.1 mm. **g** AFM image of the edge of a $MoS_2$ crystal (transferred onto $SiO_2$/Si) and its corresponding height profile. Scale bar: 1 μm. **h** Comparison of the uniform area (bars) and growth time (green squares) of monolayer $MoS_2$ on glass and other monolayer TMDs on various substrates reported in the literatures[14,26,29,38-44]

line with $Mo^{4+}$ (Supplementary Fig. 2), and S $2p_{3/2}$ (at 162.2 eV) and $2p_{1/2}$ (at 163.4 eV) peaks are assigned to $S^{2-}$. These results agree well with the standard XPS data for $MoS_2$[18], thus suggesting the formation of a $MoS_2$ film. Moreover, the typical optical microscopy (OM) of the sample surface shows a highly uniform color contrast, indicating the homogenous thickness and in-plane continuity of the $MoS_2$ film (Fig. 1c).

The layer thickness and crystal quality of the achieved $MoS_2$ film were then determined by Raman spectroscopy and photoluminescence (PL) analyses. As shown in Fig. 1d, the randomly selected Raman spectra (collected from the 100 sampling points on the $MoS_2$-covered glass) exhibit two characteristic peaks of $MoS_2$, corresponding to the $E^1_{2g}$ (at 385.7 $cm^{-1}$) and $A_{1g}$ (at 406.2 $cm^{-1}$) vibration modes, respectively. The specific frequency difference (Δ) is 20.5 $cm^{-1}$, highly indicative of the monolayer nature of the obtained film[34]. Moreover, the typical PL spectrum presents a sharp excitonic $A$ peak at 1.88 eV (660 nm), its relatively high intensity and narrow full-width at half-maximum (54 meV) confirm the rather high crystallinity of our CVD-derived monolayer $MoS_2$[35]. Moreover, Raman mapping on the intensity of $A_{1g}$ mode manifests an extra uniform color contrast, further demonstrating its perfect thickness uniformity and good crystallinity over a large scale (Fig. 1e).

In order to investigate the intrinsic crystal quality, the growth time was deliberately reduced to 5 min. Large triangular domains are observed to be evenly dispersed on the surface by OM images,

presenting an average edge length of ~200 μm (Fig. 1f and Supplementary Fig. 3). In this regard, the continuous film is proposed to arise from the merging of composite domains, either by extending the growth time or by increasing the precursors feeding rate. Notably, in a continuous monolayer film, the average domain size obtained in this work is much larger than previously reported ones synthesized on $SiO_2$/Si[26,28] or on Au foils[36].

Interestingly, the domain size of the triangular $MoS_2$ crystals is tunable by controlling the gap distance between the Mo foil precursor and the glass substrate. When the gap is decreased from 50 to 10 mm, the nucleation density reduces dramatically, and the average edge length of the resulting domains increases from 1 to 400 μm. This phenomenon is attributed to the gradual reduction in sulfur concentration over the gap region, which suppresses the nucleation density and increases the domain size accordingly. However, when the gap is further reduced to 2 mm, irregularly shaped $MoO_xS_{2-x}$ crystals appear on the sample surface due to an insufficient feeding of the S precursor, similar to the observation from the published reference[37] (Supplementary Fig. 4). In addition, the $MoS_2$ domains also increase in size with increasing concentration of $O_2$ (from 1 to 6 sccm), due to the gradually increased oxidization of the Mo foil and sufficient supply of the metal precursor (Supplementary Fig. 5).

In order to accurately determine the thickness of the $MoS_2$ film, a modified polymethyl methacrylate (PMMA)-assisted method was also developed to transfer the as-grown $MoS_2$ film

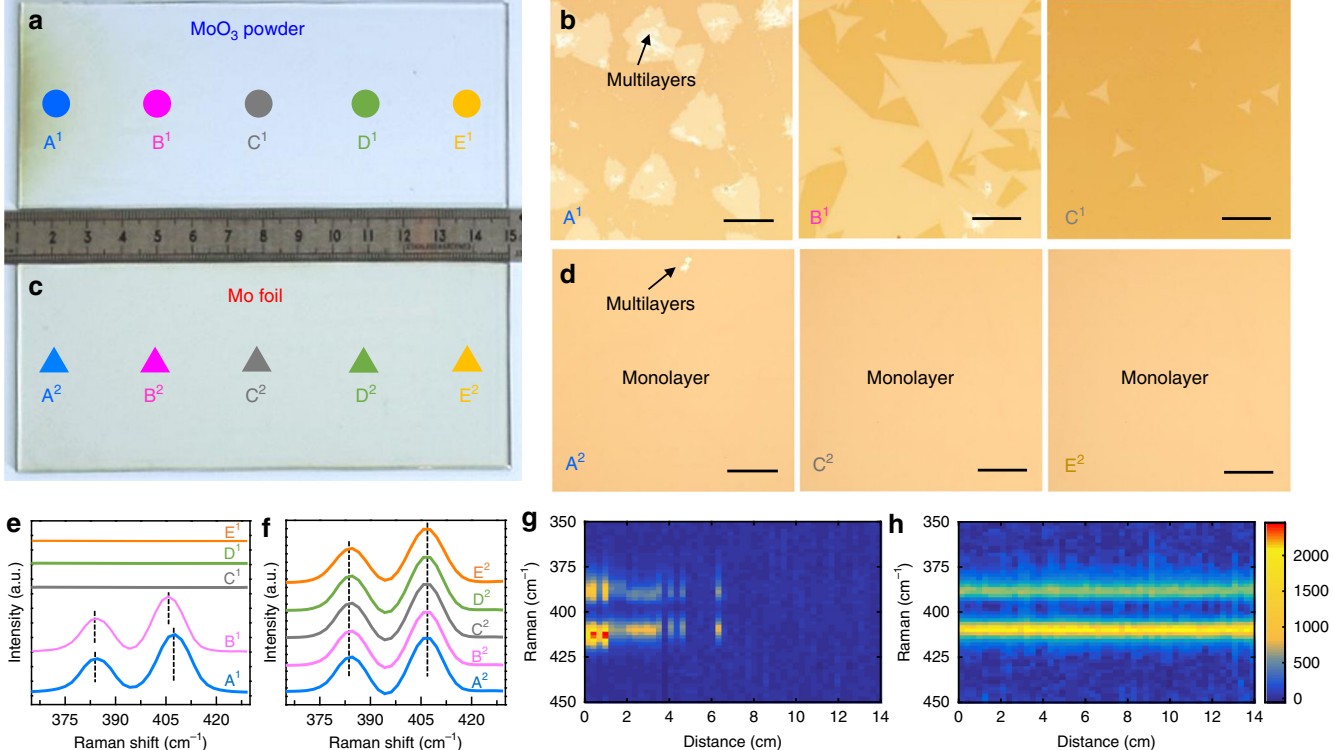

**Fig. 2** Comparison of MoS$_2$ synthesized by a point-to-face and face-to-face metal-precursor supply routes. **a**, **c** Photographs of MoS$_2$ growth on 6-inch soda-lime glass using **a** MoO$_3$ powder and **c** Mo foil as precursors. The gas-flow direction was from left to right in both cases. **b**, **d** Corresponding OM images of MoS$_2$ synthesized using **b** MoO$_3$ powder and **d** Mo foil as precursors, at the points labeled A$^1$, B$^1$, and C$^1$ in **a** and A$^2$, C$^2$, and E$^2$ in **c** (with the different locations marked by colored letters; scale bars: 100 μm). **e**, **f** Corresponding Raman spectra of MoS$_2$ synthesized with the MoO$_3$ powder and the Mo foil precursors, respectively, at the points labeled A–E in **a** and **c**. **g**, **h** Color-coded images of the typical Raman modes for the samples shown in **a** and **c**, respectively, collected from 70 positions (with an interval of 2 mm along the horizontal direction)

onto the SiO$_2$/Si substrate, which will be discussed later. Figure 1g shows an atomic force microscopy (AFM) image of a MoS$_2$ domain edge, and the corresponding height profile reveals a value of 0.77 nm, the same as that of the previously reported data for a monolayer thickness[20]. Notably, using the current face-to-face metal-precursor feeding route, the thickness uniform region is much larger than that of previously reported ones using the pre-deposited and point-to-face metal-precursor supply methods (Fig. 1e)[14,26,29,38–44]. For the batch production capability, the maximum sample size is only limited by the diameter of the tube furnace (3-inch diameter for the current experiment); larger sample size is attainable by increasing the size of the furnace.

To highlight the excellent thickness uniformity of the 6-inch monolayer MoS$_2$ film on glass synthesized through a face-to-face metal-precursor feeding method, a point-to-face feeding route with the MoO$_3$ powder precursor was also utilized to grow MoS$_2$ on glass for comparison. Digital photographs of two pieces of 6-inch MoS$_2$/glass samples, synthesized with the MoO$_3$ powder placed upstream and the Mo foil placed on top, are displayed in Fig. 2a and c, respectively. Notably, the different thickness uniformity of the two-type samples is visible even to the naked eye. A "point-to-face" route-derived MoS$_2$/glass shows gradually fading brown color along the gas-flow direction ($d = 1$–8 cm, where $d$ denotes the distance from the left edge of the sample shown in the photograph), and the glass surface is almost colorless at the downstream location $d = 8$–14 cm. More specifically, the corresponding OM images of the selected points A$^1$, B$^1$, and C$^1$ (marked in Fig. 2a with circles, corresponding to $d = 1$, 4, and 7 cm, respectively), present the characteristic

morphologies of multilayer domains, larger monolayer domains (edge length 300 μm), and smaller monolayer domains (edge length 40 μm), respectively.

On the other hand, nearly homogenous contrast (uniform light-brown color) is noticeable over the entire face-to-face route-derived sample (Fig. 2c). Corresponding OM images of the points marked with A$^2$, C$^2$, and (shown by triangles in Fig. 2c) for $d = 1$, 4, and 13 cm, respectively, reveal a highly uniform color contrast over the entire surface, reconfirming the centimeter-scale uniformity of the obtained monolayer MoS$_2$ films. To address the remarkably different thickness uniformity of the two-type samples, Raman spectra were also recorded at the five points (A to E) indicated in Fig. 2a, c. For a point-to-face precursor-feeding method, Δ of the two characteristic peaks ($E^1_{2g}$ and A$_{1g}$) decreases from ~22.6 to ~20.7 cm$^{-1}$ along the upstream points (positions A$^1$ and B$^1$ in Fig. 2a), consistent with the thickness change from multilayer to monolayer (also shown in the OM images). In addition, only a faint Raman signal is visible at the downstream points (positions C$^1$, D$^1$, and E$^1$ in Fig. 2a) due to the few nucleation sites.

However, the Raman spectra for a face-to-face precursor-feeding derived sample are nearly identical for various locations along the sample, with the characteristic Δ value of 20.5 cm$^{-1}$, highly indicative of its superior thickness uniformity (Fig. 2f). Moreover, Raman data from 70 typical locations (with an interval of 2 mm) were also collected in Fig. 2g and h for the two-type samples. Clearly, a "face-to-face" precursor-feeding derived sample exhibits excellent thickness uniformity in peak intensity and frequency difference (Δ of 20.6 ± 0.3 cm$^{-1}$).

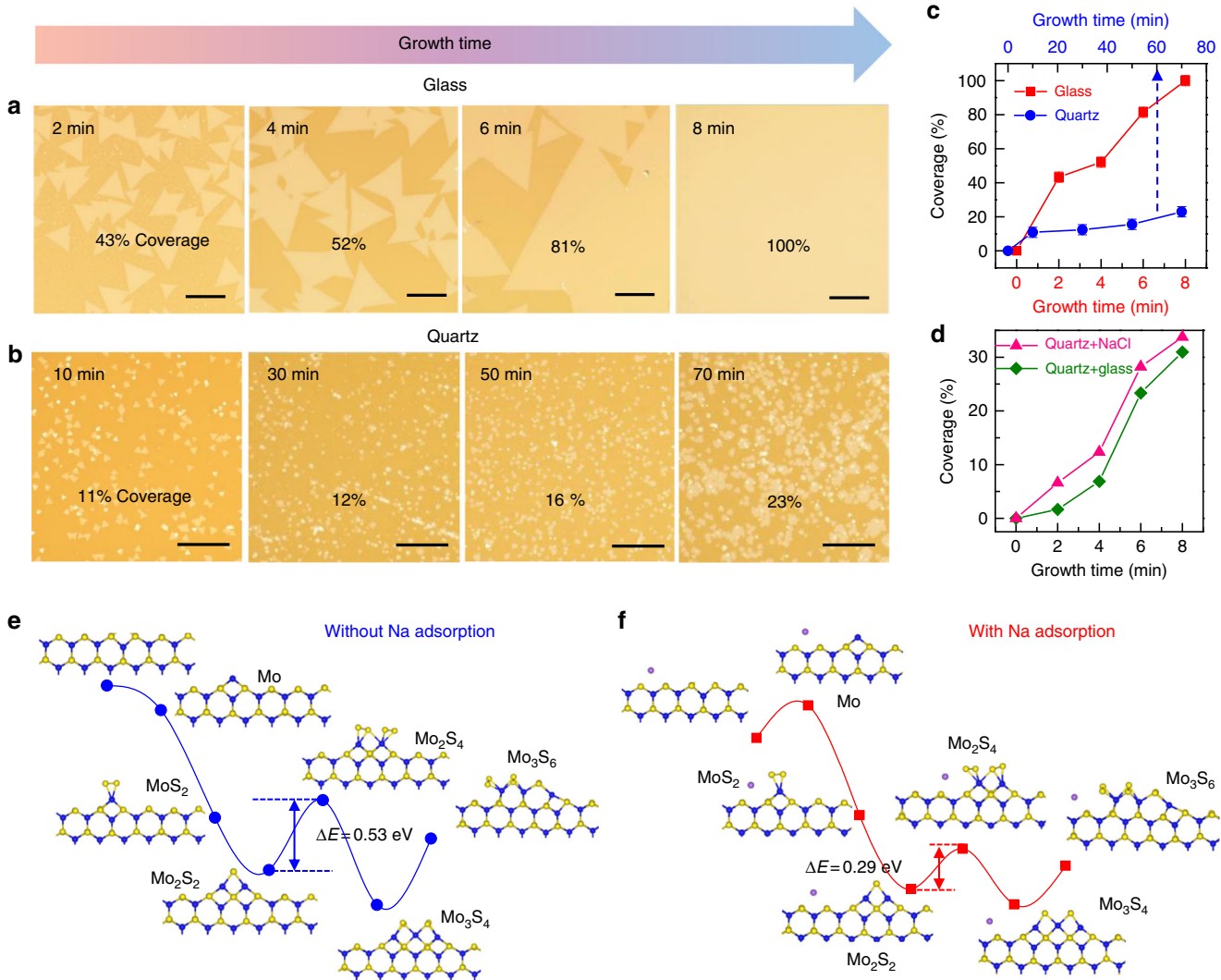

**Fig. 3** The role of Na from soda-lime glass in promoting the growth rate of monolayer $MoS_2$. Growth-time-dependent OM images of $MoS_2$ synthesized on **a** soda-lime glass with growth time from 2 to 8 min (scale bars: 100 μm) and **b** quartz glass from 10 to 70 min (scale bars: 50 μm). Other than the substrate type, all experimental parameters were identical for the samples shown in **a** and **b**. **c** $MoS_2$ coverage as a function of growth time on soda-lime glass (red) and quartz (blue). **d** $MoS_2$ coverage as a function of growth time on quartz substrates either placed downstream from the glass (green) or spin-coated with NaCl (pink) prior to growth. **e**, **f** DFT-calculated energy diagrams for $MoS_2$ growth along the S-terminated edges, **e** without and **f** with Na adsorption. The blue, yellow, and purple spheres represent molybdenum, sulfur, and sodium atoms, respectively

The disparate film thickness uniformities of the two-type $MoS_2$/glass samples (synthesized using a face-to-face and point-to-face metal-precursor feeding routes) can be explained by the different concentration gradients of the Mo species over the substrate surfaces. In the latter case, the $MoO_3$ powder is partially reduced by the sulfur vapor to form volatile $MoO_{3-x}$ species, which are subsequently transported downstream by the gas flow and react with sulfur[19]. Apparently, the concentration of the $MoO_{3-x}$ species gradually decreases along the gas-flow direction, leading to smaller and sparser $MoS_2$ crystals at the downstream positions. However, in our designed growth strategy, wherein the Mo foil is placed face-to-face to the glass substrate, the $MoO_{3-x}$ species arising from the oxidized Mo foil can be uniformly released into the gap between Mo foil and glass, resulting in homogenous nucleation and growth over the entire glass surface. Accordingly, a large-area uniform monolayer $MoS_2$ film is unexceptionally obtained.

**Growth mechanism of monolayer $MoS_2$ on soda-lime glass**. In our experiments, the continuous 6-inch monolayer $MoS_2$ film is usually synthesized on glass at a quite short time of 8 min, in line with an edge growth rate of around 1.2 μm s$^{-1}$. Notably, this growth rate is much faster than that on common insulating substrates, such as $SiO_2$/Si (15 min; 0.4 μm s$^{-1}$)[26] and sapphire (30 min; 0.2 μm s$^{-1}$)[24]. The glass substrate, thereby, plays a crucial role in the fast growth of monolayer $MoS_2$ films. To understand this, the $MoS_2$ growth both on soda-lime glass (mainly composed of $SiO_2$, $Na_2O$, and CaO) and quartz (mainly composed of $SiO_2$) were compared in detail. Typical OM images of $MoS_2$ grown for different time durations on soda-lime glass (2–8 min) and on quartz (10–70 min) are presented in Fig. 3a and b, respectively. This growth time difference should justify their different growth rates in general. As for the surface morphology, well-defined triangular flakes with an edge length of hundreds of microns can be noticed on glass, and the coverage is precisely tunable from 43 to 100% with increasing growth time from 2 to

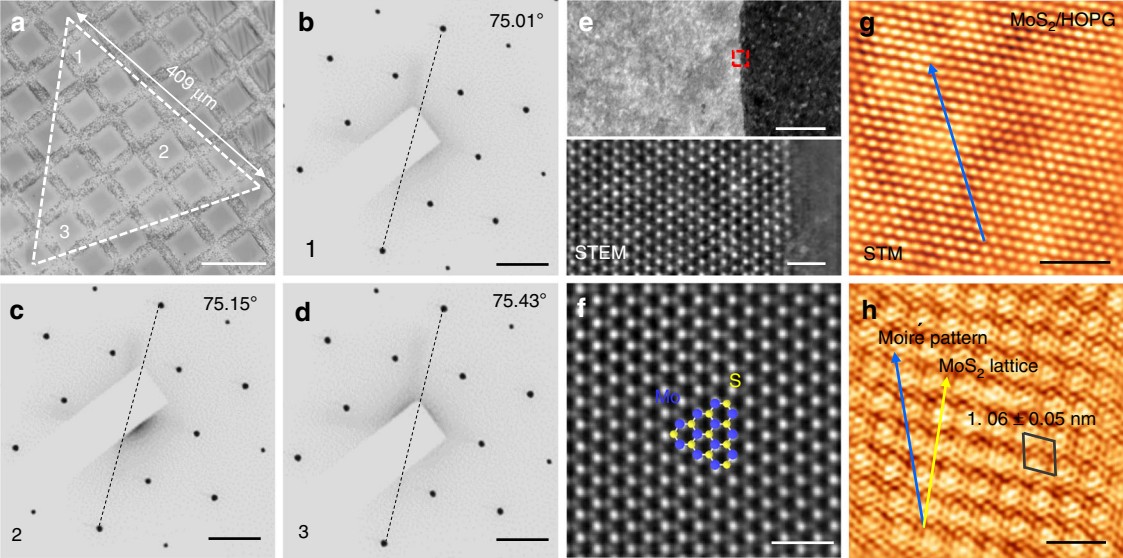

**Fig. 4** Atomic structure of transferred monolayer $MoS_2$. **a** OM image of a transferred $MoS_2$ domain on a carbon TEM grid, scale bars: 100 μm. **b–d** SAED patterns from the regions numbered 1–3 in **a**. The dashed lines indicate the rotation angles (75.01°, 75.15°, and 75.43°) with respect to the horizontal line (scale bar: 3 $nm^{-1}$). **e** STEM (upper) image of the $MoS_2$ domain edge and the enlarged view (lower) of the marked rectangular region. Scale bars: 20 and 1 nm, respectively. **f** Atomically resolved image representing the defect-free hexagonal structure of $MoS_2$. The bright spots are Mo atoms and the gray spots correspond to two-stacked S atoms. Scale bars: 1 nm. **g** Representative Moiré-scale STM image of $MoS_2$ transferred on HOPG ($V_T = -0.21$ V, $I_T = 5.19$ nA, scale bar: 5 nm) and **h** corresponding atomic-resolution STM image ($V_T = -0.33$ V, $I_T = 5.19$ nA, scale bar: 2 nm). The arrows indicate the directions of the $MoS_2$ lattice (yellow) and the Moiré pattern (blue). The unit cell for the Moiré pattern is outlined with a rhombus (period of 1.06 ± 0.05 nm)

8 min (with an interval of 2 min) (Fig. 3a, from left to right). In contrast, when using the same growth parameter, there is much poorer control of the morphology of $MoS_2$ on quartz with the growth time varying from 10 to 70 min (with an interval of 20 min) (Fig. 3b, from left to right). The sample is characterized with much smaller domain sizes (edge lengths 5–10 μm), high density of nucleation sites, and nonuniform thicknesses. Moreover, according to statistical results, the coverage of $MoS_2$/glass increases monotonically with growth time, and the growth rate is approximately 38 times higher than that on quartz (Fig. 3c; as calculated from the coverage expansion per minute). Intriguingly, by placing a piece of soda-lime glass on the upstream area of the quartz substrate, the domain size and the growth rate of $MoS_2$ on quartz are significantly increased (Supplementary Fig. 6). Considering the different chemical compositions of soda-lime glass and quartz, the metallic elements in glass, e.g., Na and Ca, should take effect in the fast growth of $MoS_2$.

To confirm this, a quartz substrate was spin-coated with a solution of $CaCl_2$ (0.01 g ml$^{-1}$) or NaCl (0.01 g ml$^{-1}$) before depositing $MoS_2$. The derived $MoS_2$ layer on the $CaCl_2$-coated quartz exhibits nonuniform thicknesses and irregular domain shapes (Supplementary Fig. 7), indicating that the effect of Ca is negligible for accelerating the growth of $MoS_2$. However, for the quartz substrate coated with NaCl, both the domain size and surface coverage of $MoS_2$ are significantly enhanced (Fig. 3d and Supplementary Fig. 6). Statistically, the average growth rate of $MoS_2$ on quartz with the assistance of upstream glass or spin-coated with NaCl is approximately 13 times faster than that on pure quartz (Fig. 3d).

In order to exclude the contribution of Cl, quartz spin-coated with NaOH solution (0.01 g ml$^{-1}$) was also employed as a substrate for $MoS_2$ growth. The result is quite similar to that for NaCl-coated substrate (Supplementary Fig. 8). Based on these

experiments, the Na element in the soda-lime glass is considered to serve as a very effective promotor in the fast growth of $MoS_2$. As an additional proof, Na is also detected in the oxidized Mo foil after $MoS_2$ growth, as evidenced by XPS spectra (Supplementary Fig. 9). Thus, Na is expected to be widely distributed in the confined space between Mo foil and glass during the CVD growth process. However, after being transferred onto $SiO_2$/Si, no Na signal appears for the monolayer $MoS_2$ film, confirming that Na only serves as an intermediate catalyst for the rapid growth of $MoS_2$ (Supplementary Fig. 9).

DFT calculations were then performed to provide an in-depth understanding of the growth mechanism. The energy diagrams for $MoS_2$ growth along the S-terminated edges were calculated with and without Na adsorption (for the first six reaction steps), as displayed in Fig. 3e and f, respectively. With the incorporation of Na, the highest energy barrier (that of the step from $Mo_2S_2$ to $Mo_2S_4$) reduces from 0.53 to 0.29 eV for $MoS_2$ growth. Assuming that the rates of atom attachment ($S_2$ and Mo) are the same for the two cases, the ratio of growth rate with Na to that without Na is estimated as $\exp[(0.53–0.29)k^{-1}T^{-1}] \approx 17$ at $T = 1000$ K. This result is in good agreement with the experimental data shown in Fig. 3c. Meanwhile, for the growth of $MoS_2$ along the Mo-terminated edge (Supplementary Fig. 10), the energy barriers are calculated to be 1.29 and 2.04 eV for those with and without Na assistance, respectively. Much faster growth of the S-terminated edge indicates the preferential evolution of Mo-terminated edges according to the published reference[27], which corroborates our experimental results (as will be discussed in Fig. 4e).

**Atomic structure characterizations of the crystal quality.** Transmission electron microscopy (TEM), scanning transmission electron microscopy (STEM), and scanning tunneling microscopy (STM) were then utilized to evaluate the crystal quality of the

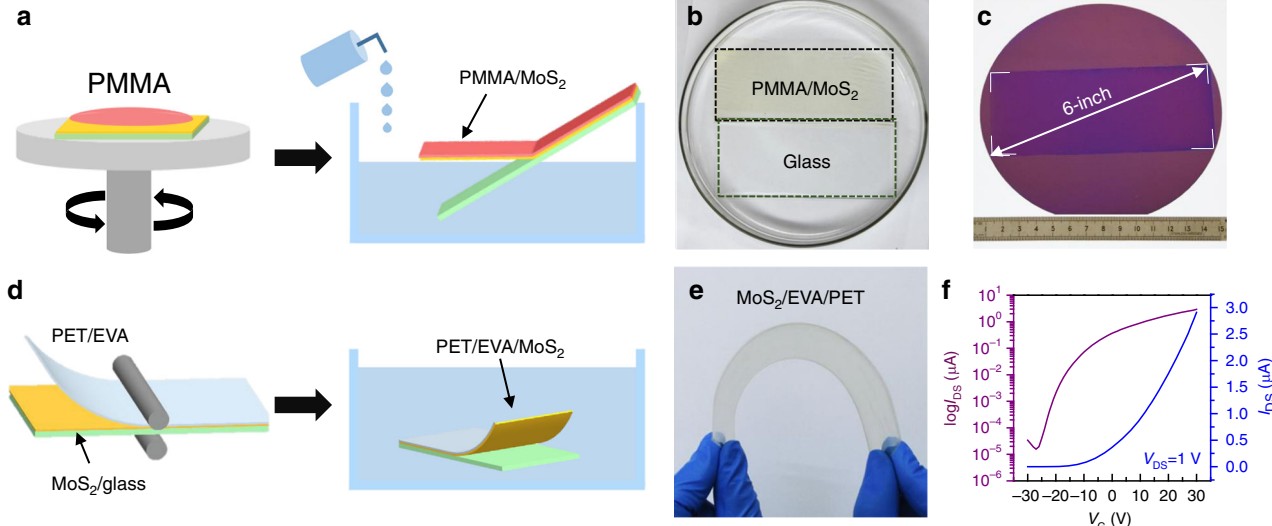

**Fig. 5** Green transfer process and device performance of monolayer $MoS_2$. **a** Schematic diagram of the PMMA-assisted etching-free transfer process onto rigid substrates. **b** Photograph of the delamination process for removing the $PMMA/MoS_2$ film from the glass substrate. **c** Photograph of the 6-inch $MoS_2$ film transferred onto a $SiO_2/Si$ substrate. **d** Schematic diagram of the roll-to-roll transfer route onto flexible substrates. **e** Photograph of a typical 6-inch uniform, flexible, monolayer $MoS_2/EVA/PET$ stack. **f** $I_{DS}$–$V_G$ curves of a typical $MoS_2$ FET device at $V_{DS} = 1\,V$

CVD-derived monolayer $MoS_2$. Figure 4a shows an OM image of a typical triangular $MoS_2$ domain with an edge length of 409 μm. Selected area electron diffraction (SAED) patterns regarding the numbered regions (1, 2, and 3 in Fig. 4a) present nearly identical lattice orientations (deviation smaller than ± 0.5°), suggesting the single-crystal nature of the $MoS_2$ triangular domain (Fig. 4b–d). Moreover, the atomically resolved image from the domain edge presents a Mo-terminated zigzag edge type (Fig. 4e), which agrees well with the previously published result[28]. Specifically, the lattice inside the domain presents a well-organized honeycomb lattice with an interatomic distance of approximately 0.32 nm (Fig. 4f), which is identical to that previously documented for $MoS_2$[45]. And the nearly defect-free structure confirms the rather high crystal quality of the CVD-derived $MoS_2$ on glass.

In order to present the relatively high stability and perfect crystal quality, the monolayer $MoS_2$ flake was transferred onto a highly oriented pyrolytic graphite (HOPG) substrate for STM characterizations. Large-area uniform, hexagonal Moiré patterns are universally observed to show a period of $1.06 \pm 0.05$ nm (Fig. 4g, as marked by a rhombus in Fig. 4h), which arises from the lattice mismatch between $MoS_2$ ($a_1 = 0.312$ nm) and graphite ($a_2 = 0.246$ nm). Briefly, STEM and STM observations provide consistent proof of the relatively high crystal quality of CVD-derived monolayer $MoS_2$, as well as its perfect stability during the harsh sample transfer process.

**Green transfer and electronic property characterizations**. To fulfill the applications of large-area uniform monolayer $MoS_2$, a convenient transfer process is highly desirable. To date, two predominant strategies using PMMA as the supporting layer have been demonstrated, i.e., electrochemical bubbling and wet chemical etching methods. The former is mainly used to transfer monolayer TMDs on conductive substrates, such as Au foils, by virtue of the weak interfacial interaction[14,29]. In the latter case, structural damage and performance degradation usually occur due to the use of acidic and alkaline solutions[44]. To mitigate this issue, facile/rapid, environmentally friendly transfer routes need to be developed for specific growth systems. Recently, Gurarslan et al.[46] and Xu et al.[47] have developed similar methods to transfer

$MoS_2$ and $WS_2$ from sapphire, based on the hydrophilic behavior of the sapphire substrate and the accompanying capillary force. For the former work, the polymer/$MoS_2$ stack needed to be picked up from the water droplet and then transferred onto target substrates, which can hardly ensure the large-scale continuity of the large-size samples. For the latter work, a NaOH solution pre-etching process was highly necessary to open a gap between polymer and sapphire, which inevitably caused damage to both sample and substrate.

In this work, by virtue of the hydrophilicity of the glass substrate, an etching-free, easy processing, and scalable transfer strategy was developed for the high-quality transfer of large-area uniform samples (Supplementary Fig. 11). As illustrated in Figs. 5a and d, both PMMA-assisted and the roll-to-roll transfer strategies were designed to transfer $MoS_2$ from glass to $SiO_2/Si$ and ethylene vinyl acetate/polyethylene terephthalate (EVA/PET) substrates, respectively. For the transfer to rigid substrates, a modified PMMA-assisted transfer process involving three essential steps was developed: first, spin-coating PMMA on $MoS_2$/glass; then making cracks along the edge of PMMA/$MoS_2$ with a knife, inducing natural penetration of water into the interface between $MoS_2$/PMMA and substrate due to their different surface energies (Supplementary Fig. 11); finally, inclining the PMMA/$MoS_2$/glass inside a container and injecting water at an optimized rate of $10\,ml\,s^{-1}$ to minimize the interface stress. In this way, the 6-inch PMMA/$MoS_2$ film can be easily delaminated from the glass substrate, mainly relying on the interfacial capillary force (Fig. 5b). Finally, the PMMA/$MoS_2$ film was fished by the target substrate and then immersed in acetone to remove PMMA (Fig. 5c). Interestingly, the entire peel-off process can be completed within 60 s for a 6-inch $MoS_2$ film. Directly injecting water rather than gradually feeding the sample into water is much easier to handle and is more suitable for transferring large-area samples.

Compared to the PMMA-assisted transfer route, the developed roll-to-roll method was more convenient as follows: adhesion of EVA/PET plastic onto $MoS_2$/glass via a portable hot lamination process, which is also used to transfer graphene[48]; immersion of the EVA/PET/$MoS_2$/glass into the deionized water, thus induces

automatically penetration of water into the interface between PET/EVA/MoS$_2$ and the glass substrate, followed with effective delamination within 5 min for a 6-inch sample (Fig. 5e).

Notably, both the PMMA-assisted and roll-to-roll transfer pathways are free of acid or alkaline etching, thus avoiding unnecessary contamination and irreversible damage, either to the MoS$_2$ film or to the glass substrate. Accordingly, these carefully designed transfer routes can preserve the original high quality of the CVD-derived monolayer MoS$_2$, and they are more environmentally friendly, efficient, and easier to operate than the commonly used wet chemical etching route. This intact transfer route also guarantees the recyclable use of the glass substrate, as presented in Supplementary Fig. 12 by Raman spectra and OM images. Even so, the soda-lime glass substrate should be used for less than three times considering the gradually decreased Na content on the glass surface.

To evaluate the electrical performance of MoS$_2$ samples, back-gated field-effect transistors (FETs) were fabricated on 300 nm SiO$_2$/Si substrates based on transferred MoS$_2$ monolayers with 10 nm Ti/50 nm Au as source and drain electrodes (Supplementary Fig. 13). The transfer characteristic curve of a typical FET device is presented in Fig. 5f, with a channel length of 1 μm and a channel width of 3 μm, which shows a typical n-type behavior. In addition, the transport characteristics of 46 randomly selected MoS$_2$ FET devices are also analyzed (Supplementary Fig. 13). The achieved mobility and on/off ratio of these devices falls in a narrow range of 6.3 to 11.4 cm$^2$ V$^{-1}$ s$^{-1}$ and 10$^5$ to 10$^6$, respectively, suggesting the relative high-quality uniformity of our large-scale MoS$_2$ samples. Both parameters are comparable to those of back-gated FETs fabricated with CVD-grown MoS$_2$ flakes[20,24,26,28,32,49–52] (Supplementary Table 1). Notably, the carrier mobility can be further improved by optimizing the interface contact by pre-treatment[24] or changing source/drain electrodes[53,54].

## Discussion

In summary, by exploiting a face-to-face metal-precursor supply method, we have successfully synthesized a 6-inch uniform monolayer MoS$_2$ film with the domain size larger than 400 μm on solid soda-lime glass. Particularly, the sample size is even scalable by increasing the size of the growth chamber, considering the rather homogenous precursor feeding route designed in this work. Intriguingly, the uniformly distributed Na in soda-lime glass is confirmed to serve as perfect catalysts for the rapid and large-scale uniform growth of monolayer MoS$_2$, according to both experimental facts and DFT calculations. By exploiting the hydrophilicity feature of glass, we have also developed an etching-free method to transfer large-area MoS$_2$ films onto targeted substrates. We believe that this work should pave the way for the cost-effective batch production, environment-friendly transfer, as well as versatile applications in both fundamental and industrial aspects.

## Methods

**Face-to-face metal-precursor feeding assisted synthesis of macroscopic uniform monolayer MoS$_2$ on soda-lime glass**. Before synthesis, the commercial soda-lime glass (14 × 6 cm$^2$; 2 mm thick) was cleaned in deionized water to remove surface impurities. A rectangular piece of Mo foil (99.95%; 0.025 mm thick) was folded into a bridge and placed on top of the soda-lime glass, as shown in Fig. 1a. The Mo foil and glass were placed in a graphite boat and loaded into a 3-inch CVD chamber. Another quartz boat containing sulfur powder (99.5%) was placed upstream, 40 cm away from the substrate. Before heating, the furnace system was purged with 80 sccm Ar for 10 min. Then, Ar (50 sccm) and O$_2$ (6 sccm) gas flows were introduced into the system to create a stable growth atmosphere. The sulfur zone and the substrate zone were heated to 100 and 720 °C within 35 min, respectively. The growth was performed for 8 min to achieve the 6-inch uniform monolayer MoS$_2$ film, and the furnace was then cooled to room temperature automatically.

**PMMA-assisted transfer of 6-inch monolayer MoS$_2$ film**. Firstly, the as-grown MoS$_2$/glass was spin-coated with PMMA (950 K, Allresist, AR-P 679.04) at a speed of 80 rpm for 1 min followed by curing at 80 °C for 10 min. Then, the edge of PMMA film was scored with a knife to provide a path for the water penetration. Next, the PMMA/MoS$_2$/glass was inclined against the edge of a petri dish, and water was injected with an optimized rate of 10 ml s$^{-1}$, as shown in Fig. 5a. In this way, the PMMA/MoS$_2$ stack was spontaneously delaminated from glass, and it was then collected by a target substrate. Finally, the PMMA coating was removed using acetone and dried under the flowing N$_2$ gas. For TEM characterizations, the PMMA layer was dissolved with acetone droplets and dried naturally.

**Roll-to-roll transfer of monolayer MoS$_2$ film on PET**. The 6-inch uniform MoS$_2$ film on soda-lime glass was hot laminated at 180 °C with EVA (50 μm thick) pre-coated with PET (75 μm thick) using a lamination machine to form a glass/MoS$_2$/EVA/PET stack. Then, the glass/MoS$_2$/EVA/PET film was soaked in deionized water for ~5 min. After delamination, the MoS$_2$/EVA/PET stack was dried by flowing N$_2$.

**DFT calculations**. All calculations were performed adopting the Perdew–Burke–Ernzerhof[55] parametrization for the generalized gradient approximation of the exchange-correlation effect embedded in the Vienna Ab initio Simulation Package[56,57]. The ion–electron interaction was described by the projector-augmented wave potentials[58,59]. Following the kink-flow scheme using a nanoreactor model for graphene[60], S$_2$ and Mo were introduced stepwise to the S- or Mo-terminated growth front. The growth fronts were modeled by zigzag nanoribbons of nine lattice units along the periodic direction. The energies of these nanoribbons with and without Na adsorption were taken as references for constructing the energy diagrams. To simplify the discussion, the chemical potential of S was obtained for the most stable sulfur molecule S$_2$ at the growth temperature of 1000 K and partial pressure of 10 kPa, considering their transitional, rotational, and vibrational contributions[61]. Under such assumptions, the chemical potential of S$_2$ was calculated to be about −10 eV, while those of the bulk material were used for Mo and Na. It should be noted that the choices of chemical potentials did not influence the relative energies for the two cases with and without Na adsorption. The adsorption energies of Na on Mo- and S-terminated edges were calculated as −0.82 and −2.09 eV, indicating their favorable incorporation into the growth fronts.

**Characterization of MoS$_2$ films**. OM (Nikon ECLIPSE, LV100ND), contact angle measuring system (Dataphysics, OCA20), Raman spectroscopy (HORIBA, Lab-RAM HR-800, with an excitation wavelength of 514 nm), AFM (Bruker, Dimension Icon), XPS (Kratos Analytical AXIS-Ultra with monochromatic Al Kα X-ray), TEM (JEOL JEM-2100F), STEM (Titan Cubed Themis G2), and ultra high vacuum low temperature (UHV LT) STM were used to characterize the optical and structural properties of the MoS$_2$ sample.

**Data availability**. The data reported by this article are available from the corresponding author upon reasonable request.

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

## Acknowledgements

This work was supported by the National Natural Science Foundation of China (Nos. 51290272, 51472008, 61774003, and 11327902), the National Key Research and Development Program of China (Nos. 2016YFA0200103, 2017YFA0304600, and 2017YFA0205700), the Open Research Fund Program of the State Key Laboratory of Low-Dimensional Quantum Physics (No. KF201601), the Beijing Municipal Science and Technology Commission (No. Z161100002116020), the Youth 1000-Talent Program of China, the Shenzhen Basic Research Project (No. JCYJ20170407155608882), and the Development and Reform Commission of Shenzhen Municipality for the development of the "Low-Dimensional Materials and Devices" Discipline. The authors also acknowledge the Electron Microscopy Laboratory in Peking University for the use of Cs corrected electron microscope.

## Author contributions

Y.Z. proposed and supervised the project; P.Y. and Y.Z. designed the experiments; P.Y. performed the large-area uniform MoS2 film synthesis and transfer experiments, and did the OM, Raman, AFM, and XPS measurement; Z.Z. conducted TEM measurement and prepared the STEM sample; M.H. conducted the STM characterizations and data analysis; S.C. and P.G. conducted the STEM measurement; J.S. and Q.C. fabricated devices on SiO2/Si and carried out the electrical properties measurement; L.Z. and Q.Z. did the Raman mapping measurement; X.Z. performed DFT calculations; P.Y. and Y.Z. wrote the manuscript; All the authors contributed to the results analysis and discussions.

## Additional information

**Competing interests:** The authors declare no competing interests.

