## [Peer Review File · Nature Communications]

Reviewers' Comments:

Reviewer #1:

Remarks to the Author:

This manuscript reports a direct synthesis and etching-free transfer route for 6-inch MoS₂ monolayers. The reported "face-to-face" metal precursor supply route, as well as the use of soda-lime glass as the growth substrate, are novel. In addition, the successful growth of uniform MoS₂ monolayer films and the DFT calculations describing the growth mechanism, provide novel and relevant TMD work. Therefore, I recommend this work for publication after the authors address the following comments:

1. In line 36, the following paper could be cited: Lin, Zhong, et al. "2D materials advances: from large scale synthesis and controlled heterostructures to improved characterization techniques, defects and applications." *2D Materials* 3.4 (2016): 042001.
2. In line 37, "edge catalytic activities" does not require the large area growth of TMDs.
3. In line 53, the authors should be "Lee et al.", not "Lin et al".
4. In figure 3a, the optical image contrast of 8min growth on glass is different from other images.
5. In figure S12, the authors mentioned the substrate can be reused after synthesis and transfer process. However, the Na distribution at the soda-lime glass surface can change after MoS₂ synthesis process, which can possibly affect the sample morphology. The reviewer suggests that the three consecutive growth results should be compared using the same growth conditions (provide typical optical images in Supplementary material), and the growth time should be added in figure S12.

Reviewer #2:

Remarks to the Author:

This manuscript explores methods to create large area continuous films of monolayer MoS₂. This topic is very important and hot right now and methods to create high quality layered semiconductors of monolayer thickness are critical to future development of this field. Also the methods must be scalable, rapid, and affordable. This paper meets all these requirements with a new approach to create MoS₂ monolayers. It goes beyond current state of the art and extends the field in a positive direction. I was very impressed with this paper and strongly recommend it for publication in *Nature Comm*. It is one of the best papers I have reviewed on 2D material synthesis for a long time and this is across all journals, including *Nature*.

The paper compares different synthesis approach of Mo foil and MoO₃ powder. It has complete characterization of the materials using Raman, PL, optical imaging, TEM. It then evaluates the electronic properties in devices and on flexible plastics. It has covered all the necessary elements to prove their results and therefore I would recommend to publish as is, because I cannot find faults or ways to improve this.

In summary, this is an excellent detailed piece of research that will have major impact in the field and will generate a large number of citations due to the ease of implementation and the excellent quality of materials produced. I can strongly support this work to be published and look forward to the impact this will have in the near future for scalable growth of various 2D materials.

Reviewer #3:

Remarks to the Author:

The authors report the direct synthesis of 6-inch uniform monolayer MoS₂ on the solid soda-lime glass, through a designed "face-to-face" metal-precursor supply route in a facile chemical vapour deposition process. This is an important breakthrough for CVD growth of TMD using non-epitaxial substrate. They have also carried out detailed mechanistic studies to show that the highly efficient growth is facilitated by Na catalysts.

The work deserves to be published as it is well executed and well written.

Some comments to be addressed:

1. The author should comment on the novelty of their transfer method. Both PMMA-assisted and roll-to-roll transfer strategies have been used to transfer two-dimensional materials. When compared to other works (ACS Nano 8, 11522–11528 (2014); ACS Nano 9, 6178–6187 (2015)), the present manuscript reported a similar transfer strategy only by injecting water rather than gradually feeding the sample into water.
2. The electrical property is not better than previous reports (J. Am. Chem. Soc. 2015, 137, 15632–15635). What is the limiting factor if high quality crystal is grown? The uniformity of the film need be calculated. In Figure 3a, the coverage is tunable from ~43% to ~100% with increasing growth time from 2 to 8 min.
3. Why is the density of crystals reduced with increasing growth time?
4. The authors have cited a previous work by Chen et. al.. "Chemical vapor deposition of large-size monolayer MoSe₂ crystals on molten glass. J. Am. Chem. Soc. 139, 1073–1076 (2017)" which also applied similar face-to-face evaporation using Mo foil. The authors should state explicitly that such face-to-face evaporation was also applied by this previous group, and state the modification applied by the current work, if any over the previous.

The work can be published pending revision.

Nature Communications Manuscript ID: NCOMMS-17-27195A

Title: “Batch production of 6-inch uniform monolayer MoS₂ catalyzed by sodium in glass”

Author(s): Pengfei Yang, Xiaolong Zou, Zhepeng Zhang, Min Hong, Jianping Shi, Shulin Chen, Jiapei Shu, Liyun Zhao, Shaolong Jiang, Xiebo Zhou, Yahuan Huan, Chunyu Xie, Peng Gao, Qing Chen, Qing Zhang, Zhongfan Liu, and Yanfeng Zhang

Dear editor,

Many thanks for your careful evaluation of our manuscript “*Batch production of 6-inch uniform monolayer MoS₂ catalyzed by sodium in glass*” by Pengfei Yang et al., Manuscript ID *NCOMMS-17-27195A*, intended for publication in *Nature Communications*. We also appreciate the very constructive suggestions and comments raised by the reviewers, which are really helpful for improving the quality of our work. According to the reviewers’ kind suggestions, we have carefully revised our manuscript. Below we respond point-by-point to the reviewers’ comments, along with a revision summary attached at the beginning of this reply.

With these revisions, we sincerely hope that, the manuscript could meet the requirements for publication in *Nature Communications*.

Sincerely yours,

Yanfeng Zhang, Dr. Prof.

Department of Materials Science and Engineering

College of Engineering

Peking University

Beijing 100871, P. R. China

Tel & Fax: (+86)10-6275-7157

E-mail: yanfengzhang@pku.edu.cn

Revision Summary

- 1) According to the comment by **Reviewer 1**, we have changed the citation format of nine references, from citing by corresponding author to the first author in the revised manuscript.
- 2) According to the comment by **Reviewer 1**, we have adjusted the color contrast of the OM image of 8 min growth sample to make it accordant with others in **Fig. 3a**.
- 3) According to the comment by **Reviewer 1**, we have added the typical optical images and performed XPS measurement of MoS₂ grown on glass used for six times (new **Supplementary Fig. 12**). These results reveal that, the soda-lime glass substrate should be reused for less than three times considering the decreased Na content on the glass surface.
- 4) According to the comment by **Reviewer 3** about the novelty of our transfer method, we have incorporated more detailed descriptions of the improvements of our transfer method, comparing with the cited literatures in **Page 8** in this revised manuscript.
- 5) According to the comment by **Reviewer 3**, we have carefully checked the electrical properties of CVD-grown monolayer MoS₂ through fabricating back-gated field-effect transistors, as listed in **Supplementary Table 1**. To show the electrical property uniformity, we have supplemented a statistics over 46 FET devices of the MoS₂ sample in new **Supplementary Fig. 13** in this revised manuscript.
- 6) According to the comment by **Reviewer 3**, we have analyzed the reason of the decreased nucleation density with increasing growth time, which is mainly due to the O₂ etching effect to the unsteady nucleus and small domains of MoS₂ at the initial growth process.
- 7) According to the comment by **Reviewer 3**, we have provided **Table R2** to compare the difference between our experimental setup and the literature referred by the reviewer. We have supplemented some discussion to illustrate the novelty of our face-to-face metal precursor supply route in **Page 2** and **Page 3**.
- 8) We have cited some new references (**Refs. 5, 23, 49-52**) in this revised manuscript according to the kind suggestions by **Reviewer 1** and **Reviewer 3**.

Listed below are the point-by-point reply to the reviewers' comments.

Editor (Comments to the Author):

Dear Prof Zhang,

*Your manuscript entitled "Batch production of 6-inch uniform monolayer MoS₂ catalyzed by sodium in glass" has now been seen by 3 referees. You will see from their comments below that while they find your work of interest, some important points are raised. We are interested in the possibility of publishing your study in *Nature Communications*, but would like to consider your response to these concerns in the form of a revised manuscript before we make a final decision on publication.*

We therefore invite you to revise and resubmit your manuscript, taking into account the points raised. Please highlight all changes in the manuscript text file.

*We hope to receive your revised paper within three months; please let us know if you aren't able to submit it within this time so that we can discuss how best to proceed. If we don't hear from you, and the revision process takes significantly longer, we will close your file. In this event, we will still be happy to reconsider your paper at a later date, as long as nothing similar has been accepted for publication at *Nature Communications* or published elsewhere in the meantime.*

Please do not hesitate to contact me if you have any questions or would like to discuss these revisions further. We look forward to seeing the revised manuscript and thank you for the opportunity to review your work.

Our response:

Dear Dr. Liu,

We are very thankful for your constructive advices and the kind opportunity offered to allow us to revise our manuscript.

In this revised version, we have made great efforts to address the points raised by the reviewers, by adding some detailed discussion and performing more supplementary experiments, especially for the comments from **Reviewer 1** and **Reviewer 3**.

Through these careful revisions, we sincerely hope that, our work could meet the requirement for publication in *Nature Communications*, for its important results regarding the batch production, facile transfer and application of transition metal dichalcogenides.

Reviewer #1 (Remarks to the Author):

This manuscript reports a direct synthesis and etching-free transfer route for 6-inch MoS₂ monolayers. The reported “face-to-face” metal precursor supply route, as well as the use of soda-lime glass as the growth substrate, are novel. In addition, the successful growth of uniform MoS₂ monolayer films and the DFT calculations describing the growth mechanism, provide novel and relevant TMD work. Therefore, I recommend this work for publication after the authors address the following comments:

Our response:

We are very thankful for the reviewer’s very positive evaluation on the significance of our work, especially regarding the “face-to-face” metal precursor supply route and the use of soda-lime glass substrate, and the DFT calculations about the growth mechanism. The issues raised by the reviewer are considered very carefully and addressed point-by-point as follows:

1. In line 36, the following paper could be cited: Lin, Zhong, et al. "2D materials advances: from large scale synthesis and controlled heterostructures to improved characterization techniques, defects and applications." 2D Materials 3.4 (2016): 042001.

Our response:

We are very thankful for the reviewer’s very kind suggestion. This suggested reference makes a comprehensive review on the recent advances towards the large-scale syntheses and diverse applications of two-dimensional materials, which is representative in the recent reviews of transition metal dichalcogenides (TMDs). We have cited the reference as a new **Ref. 5** in **Line 36** of the revised manuscript.

2. In line 37, “edge catalytic activities” does not require the large area growth of TMDs.

Our response:

We appreciate the reviewer’s very constructive comment and suggestion. We agree with the reviewer that, the edge catalytic property is not directly related to the desire of large-area growth of TMDs. We have accordingly revised the sentence and updated the reference in **Line 36** by “... *direct band gap semiconductors, such as MoS₂ and WS₂, exhibit ultrahigh optical responsivity⁶, efficient valley polarization^{7,8} and strong light-matter coupling⁹...*”

3. In line 53, the authors should be “Lee et al.”, not “Lin et al”.

Our response:

We are very thankful for the reviewer’s kind remind.

We have changed the total nine references format from citing by corresponding author to the first author in the revised manuscript. For instance, in **Line 53** by “...*Lee et al.* reported the successful synthesis of MoS_2 atomic layers on SiO_2/Si ...”

4. In figure 3a, the optical image contrast of 8min growth on glass is different from other images.

Our response:

We are appreciative of the reviewer’s very kind notice. The contrast difference of the optical microscopy (OM) image arises from the varying intensity of the white light brightness of the optical microscope.

According to the reviewer’s kind remind, we have adjusted the contrast of the OM image of 8 min growth sample to make it accordant with others. The new **Fig. 3a** is as follow.

Figure R1 (also shown in **Fig. 3a** in the revised manuscript) | Growth-time-dependent OM images of MoS_2 synthesized on (a) soda-lime glass with growth time varying from 2 to 8 min (scale bars: 100 μm).

5. In figure S12, the authors mentioned the substrate can be reused after synthesis and transfer process. However, the Na distribution at the soda-lime glass surface can change after MoS_2 synthesis process, which can possibly affect the sample morphology. The reviewer suggests that the three consecutive growth results should be compared using the same growth conditions (provide typical optical images in Supplementary material), and the growth time should be added in figure S12.

Our response:

We appreciate the reviewer’s very kind suggestion regarding the growth results on the glass substrate reused for more times.

We agree with the reviewer that, the Na concentration on the surface of the soda-lime glass will decrease with increasing reuse times and thus affect the sample morphology. We have supplemented the optical images of 1st to 6th growth of MoS_2 on the reused glass and added the growth conditions in the legend.

As shown in **Figure R2** (also shown in **Supplementary Fig. 12** in the revised manuscript), under the identical growth time of 3 minutes, the morphology of MoS_2 domains shows almost no distinct changes on the soda-lime glass used for three times, and the average edge length of the domains generally maintains at $\sim 80 \mu\text{m}$.

However, when the glass substrate is used for more than three times, the average edge length and thickness of MoS₂ change a lot. When the glass is used up to six times, the average edge length of MoS₂ domains reduces sharply to ~20 μm, together with the formation of a large number of multilayers domains, which is similar with that of MoS₂ growth on quartz.

According to the XPS measurements, this phenomenon is mainly due to the decreased Na content on the glass surface. Therefore, it is suggested that, the soda-lime glass can be used for no more than three times. The results also support the Na catalytic growth mechanism of MoS₂ on soda-lime glass, as proposed by our work.

We have added this discussion in **Page 9** in revised manuscript by “...Even so, the soda-lime glass substrate should be used for less than three times considering the gradually decreased Na content on the glass surface....”

Figure R2 (also shown in Supplementary Fig. 12 in the revised manuscript) | (a-c) Optical images of the growth of MoS₂ on soda-lime glass used for three times. All the other experimental conditions ($T = \sim 720$ °C, $t = \sim 3$ min) are the same. Scale bars: 100 μm. (d) Raman spectra of MoS₂ synthesized on the glass used for three times. (e-g) Optical images of the growth of MoS₂ on soda-lime glass used for four to six times with the same growth condition. Scale bars: 100 μm. (h) XPS spectra of the Na 1s peak for the soda-lime glass after MoS₂ growth for 1st and 6th times, respectively. These results indicate that, the repeatable use of the glass substrate is probable for three times in the current growth system. However, when the glass substrate is used for four to six times, the average edge length of the MoS₂ domains will decrease and the thickness will increase, due to the greatly decreased Na content on the surface of glass.

Reviewer #2 (Remarks to the Author):

This manuscript explores methods to create large area continuous films of monolayer MoS₂. This topic is very important and hot right now and methods to create high quality layered semiconductors of monolayer thickness are critical to future development of this field. Also the methods must be scalable, rapid, and affordable. This paper meets all these requirements with a new approach to create MoS₂ monolayers. It goes beyond current state of the art and extends the field in a positive direction. I was very impressed with this paper and strongly recommend it for publication in Nature Comm. It is one of the best papers I have reviewed on 2D material synthesis for a long time and this is across all journals, including Nature.

The paper compares different synthesis approach of Mo foil and MoO₃ powder. It has complete characterization of the materials using Raman, PL, optical imaging, TEM. It then evaluates the electronic properties in devices and on flexible plastics. It has covered all the necessary elements to prove their results and therefore I would recommend to publish as is, because I cannot find faults or ways to improve this.

In summary, this is an excellent detailed piece of research that will have major impact in the field and will generate a large number of citations due to the ease of implementation and the excellent quality of materials produced. I can strongly support this work to be published and look forward to the impact this will have in the near future for scalable growth of various 2D materials.

Our response:

We are very grateful for the reviewer's very high evaluation on the novelty and significance of our work, especially regarding the scalable, rapid and affordable traits of our growth method. We sincerely hope that our work will boost the batch production and practical application of 2D materials.

Reviewer #3 (Remarks to the Author):

The authors report the direct synthesis of 6-inch uniform monolayer MoS₂ on the solid soda-lime glass, through a designed "face-to-face" metal-precursor supply route in a facile chemical vapour deposition process. This is an important breakthrough for CVD growth of TMD using non-epitaxial substrate. They have also carried out detailed mechanistic studies to show that the highly efficient growth is facilitated by Na catalysts. The work can be published pending revision.

Our response:

We are very grateful for the reviewer's very positive evaluation on the novelty and the significance of our work, especially regarding the successful growth of large-area uniform MoS₂ film on the non-epitaxial substrate. The issues raised by the reviewer are considered very carefully and addressed point-by-point as follows.

1. The author should comment on the novelty of their transfer method. Both PMMA-assisted and roll-to-roll transfer strategies have been used to transfer two-dimensional materials. When compared to other works (ACS Nano 8, 11522–11528 (2014); ACS Nano 9, 6178–6187 (2015)), the present manuscript reported a similar transfer strategy only by injecting water rather than gradually feeding the sample into water.

Our response:

We are appreciative of the reviewer's very valuable comment and suggestion.

We agree with the reviewer that, our modified etching-free transfer method relies on the capillary force between hydrophilic soda-lime glass and the more hydrophobic PMMA film, hereby the basic principle of our transfer strategy is identical with the two pioneer references reported by Gurarslan *et al.*, ACS Nano **8**, 11522–11528 (2014) and Xu *et al.*, ACS Nano **9**, 6178–6187 (2015).

However, there are obvious improvements of the detailed transferred process used in this work comparing with the published ones. In this work, large area uniform transfer of monolayer MoS₂ films was realized with the sample up to 10 cm × 6 cm. It is thus more scalable, easy handle and practical.

For Gurarslan *et al.*' work, they first dropped a water droplet onto the MoS₂ sample coated with polymer, then poked the edge and made cracks to initiate the water penetration, picked up the polymer/MoS₂ stack after it floating, and finally transferred it onto other substrates. This process is hard to control the peeling process, and not suitable for the transfer of large scale sample (eg. more than 1cm). In contrast, the delamination process in our work is completed by inclining the PMMA/MoS₂/glass in a water container and injecting water into the container very slowly, and the transferred MoS₂ film could maintain its continuous morphology. Specifically, this route is easy scalable to large-area samples (e.g. 6 inch). Therefore, our transfer method is facile, scalable and highly efficient.

In Xu *et al.*' work, the polymer/WS₂/sapphire need to be pre-etched in the NaOH solution for opening a gap between polymer and sapphire. The etching time and temperature of the NaOH solution need to be precisely controlled to avoid the unrecoverable damage to sample and substrate. In contrast, the route developed in our work avoids the use of any etching agent, and only by making cracks along the sample edge to facilitate the natural penetration of water. Accordingly, the original high-quality/continuity of the MoS₂ film can be well preserved.

In addition, the reported delamination process was triggered by gradually feeding the sample into water. In this process, the feeding rate needs to be maintained strictly to avoid thin polymer tearing especially for large-area samples, and it is hard to control only by manual operation. However, as presented by our work, the sample was hold stationary and inclined, and water was injected slowly into the sample-transfer container. The process is much easier to handle, and could largely decrease the damage to sample introduced by the human factor.

Following the reviewer's suggestion, we have clarified the discussion of the novelty of our transfer method and copy it here for your reference (**Page 8**):

"...Recently, Gurarslan et al.⁴⁶ and Xu et al.⁴⁷ have developed similar methods to transfer MoS₂ and WS₂ from sapphire, based on the hydrophilic behaviour of the sapphire substrate and the accompanying capillary force. For the former work, the polymer/MoS₂ stack needed to be picked up from the water droplet and then transferred onto target substrates, which can hardly ensure the large-scale continuity of the large-size samples. For the latter work, a NaOH solution pre-etching process was highly necessary to open a gap between polymer and sapphire, which inevitably caused damage to both sample and substrate.

In this work, by virtue of the hydrophilicity of the glass substrate, an etching-free, easy processing and scalable transfer strategy was developed for the high quality transfer of large-area uniform samples (Supplementary Fig. 11)...."

2. *The electrical property is not better than previous reports (J. Am. Chem. Soc. 2015, 137, 15632–15635). What is the limiting factor if high quality crystal is grown? The uniformity of the film need be calculated. In Figure 3a, the coverage is tunable from ~43% to ~100% with increasing growth time from 2 to 8 min.*

Our response:

We are very thankful for the reviewer's very constructive comment. We have looked over the electrical properties and fabrication methods of back-gated field-effect transistors (FETs) based on CVD-grown monolayer MoS₂ flakes as published previously (as listed in **Table R1**, also shown in **Supplementary Table 1** in the revised manuscript). We agree that, the electrical properties of our devices are comparable with most of the previous literatures, but not better than the reference suggested by the reviewer.

We notice that, there is an obvious Schottky barrier existing in our devices, as evidenced by the nonlinear shape of the output characteristics (I_{DS} - V_{DS}) from negative to positive V_{DS} range (**Supplementary Fig. 13**). Some references have reported that, the contact resistance (derived from Schottky barrier) had significant effect on the charge transport properties (Schmidt, H. *et al.*, *Nano Lett.* **14**, 1909–1913 (2014), Liu, H. *et al.*, *Nano Lett.* **13**, 2640 (2013)).

To decrease the contact resistance and achieve an optimal device performance, Radisavljevic, B *et al.* annealed the MoS₂ FETs at 200 °C, under argon and hydrogen flow for 2h. After annealing, devices showed a factor of 10 resistance decrease (*Nat. Nanotechnol.* **6**, 147–150 (2011)). Schmidt, H. *et al.* employed a two-step annealing process: first at 200°C for 2 h in N₂ and subsequently at 120°C for 4–10 h in vacuum. They found that, the second annealing process had a significant effect on the increase of the carrier concentration (as large as $7 \times 10^{12} \text{ cm}^{-2}$), which was caused by the removal of adsorbents such as O₂ or H₂O, as is known to deplete negative charge carriers (*Nano Lett.*, **14**, 1909–1913 (2014)). Liu, B. *et al.* also reported that, the MoS₂ back-gated devices showed Ohmic contact after vacuum annealing (*ACS Nano.* **8**, 5304 (2014)). In the work of Chen *et al.*, the fabricated FET devices were annealed at 450 °C for 4 h in an argon and hydrogen mixture in the vacuum for a better contact before electrical measurements (*J. Am. Chem. Soc.* **137**, 15632–15635 (2015)).

In summary, pretreatment of the MoS₂ FET devices by annealing in vacuum is an effective method to reduce the contact resistance and improve the electrical property. Besides, selecting low work function metals like Sc as electrode can also help to achieve higher carrier injection and lower contact resistance (Das, S. *et al.*, *Nano Lett.* **13**, 100–105 (2013)).

Therefore, we believe our electrical transport properties can be further improved by interface engineering, either by annealing the devices or changing the electrodes. Accordingly, we have revised the manuscript, and the relevant references were also supplemented as new **Refs. 49–52**.

“...Both parameters are comparable to those of back-gated FETs fabricated with CVD-grown MoS₂ flakes^{20,24,26,28,32,49–52} (Supplementary Table S1). Notably, the carrier mobility can be further improved by optimizing the interface contact by pretreatment²⁴ or changing source/drain electrodes^{53,54}....”

CVD-grown monolayer MoS ₂	Mobility (cm ² V ⁻¹ s ⁻¹)	ON/OFF ratio	Method	Metal electrodes	Channel length and width	Pretreatment	Ref.
MoS ₂ /SiO ₂	0.02	10 ⁴	photolithography	Au	L=14 μm, W=60 μm	—	Adv. Mater. , 24 , 2320–2325 (2012)
MoS ₂ /SiO ₂	0.2–3	10 ⁴ –10 ⁶	EBL	5 nm Ti/50 nm Au	L=1 μm,	unannealed	ACS Nano , 8 , 5304–5314 (2014)
MoS ₂ /SiO ₂	1.2	10 ⁷	EBL	5 nm Ti/50 nm Au	L=1 μm	—	Nano Lett. 13 , 1852–1857 (2013)
MoS ₂ /SiO ₂	1–8	10 ⁵ –10 ⁷	EBL	50 nm Al/5 nm Cr/50 nm Au	—	unannealed	Nat. Mater. 12 , 554–561 (2013)
MoS ₂ /SiO ₂	10	10 ⁶	Lithography and reaction ion etching	3 nm Ti/50 nm Au	L=100 μm, W=10 μm	—	Nat. Mater. 12 , 754–9 (2013)
MoS ₂ /glass	6.3–11.4	10 ⁵ –10 ⁷	EBL	10 nm Ti/50 nm Au	L=1 μm, W=3 μm	unannealed	This work
MoS ₂ /SiO ₂	8.2–11.4	10 ⁶	EBL	1 nm Cr/30 nm Au	—	annealed at 150 °C for 1h	Nat. Commun. 6 , 6128 (2015)
MoS ₂ /glass	3–15	10 ⁶ –10 ⁸	EBL	15 nm Cr/50 nm Au	L=8 μm, W=3 μm	unannealed	Chem. Mater. , 29 , 6095–6103 (2017)
MoS ₂ /SiO ₂	64	10 ⁷	EBL and oxygen plasma etching	30 nm Ni/20 nm Au	L=1,2,4,8,30 μm, W=4 μm	unannealed	Adv. Funct. Mater. , 1605896 (2017)
MoS ₂ /sapphire	90	10 ⁷	UV-photolithography	5 nm Ti/ 30 nm Au	L=20 μm, W=10.7 μm	annealed at 450 °C for 4h	J. Am. Chem. Soc. 137 , 15632–15635 (2015).

Table R1 (also shown in Supplementary Table 1 in the revised manuscript) | Comparison of the electronic properties of back-gated FET devices fabricated with CVD-grown monolayer MoS₂ samples (measured at room temperature).

Following the reviewer’s comment on the electrical property uniformity of our samples, we have measured the electrical properties of 46 randomly picked FET devices from the same sample, and the statistical result is presented in **Figure R5** (also shown in **Supplementary Fig. 13c** in the revised manuscript). The mobility data fell in a relative narrow range of 6.3–11.4 cm² V⁻¹ s⁻¹, suggesting the high quality uniformity of the transferred MoS₂ monolayer. We have added this information into the manuscript and copied it here for your check.

“... In addition, the transport characteristics of 46 randomly selected MoS₂ FET devices are also analyzed (Supplementary Fig. 13). The achieved mobility and on/off ratio of these devices falls in a narrow range of 6.3–11.4 cm² V⁻¹ s⁻¹ and 10⁵–10⁶, respectively, suggesting the relative high quality uniformity of our large-scale MoS₂ samples. Both parameters are comparable to those of back-gated FETs fabricated with CVD-grown MoS₂ flakes^{18,20,25,20,24,26,28,32,49-52} (Supplementary Table 1). Notably, the carrier mobility can be further improved by optimizing the interface contact by pretreatment²⁴ or changing source/drain electrodes^{53,54}....”

Figure R5 (also shown in Supplementary Fig. 13c in the revised manuscript) | A summary of the carrier mobility and the corresponding on/off ratio of 46 MoS₂ FET devices.

3. Why is the density of crystals reduced with increasing growth time?

Our response:

We are very thankful for the reviewer’s very constructive comment.

As described in our manuscript, O₂ is mixed with Ar as carrier gas in our growth process. Notably, the O₂ carrier is not only introduced to oxidize the Mo foil so as to release MoO_{3-x} species, but also as an etching agent for removing the small domains appeared in the CVD process.

When excess O₂ is introduced, the etching effect to the evolved MoS₂ flakes is more obvious, as seen by their small domain sizes or by the formation of Mo oxides (**Supplementary Figure 5**, achieved under the similar condition). With increasing growth time, unsteady nucleus and small domains were usually etched off by O₂. Therefore, the nucleation density of MoS₂ was a little bit decreased, ensuring the growth of large domain MoS₂ flakes or complete films. Chen, W. *et al.* J. Am. Chem. Soc. **137**, 15632–15635 (2015), also reported the etching effect of O₂.

4. The authors have cited a previous work by Chen *et al.* “Chemical vapor deposition of large-size monolayer MoSe₂ crystals on molten glass. J. Am. Chem. Soc. 139, 1073–1076 (2017)” which also applied similar face-to-face evaporation using Mo foil. The authors should state explicitly that such face-to-face evaporation was also applied by this previous group, and state the modification applied by the current work, if any over the previous.

Our response:

We are grateful for the reviewer’s very valuable suggestion. As mentioned by the reviewer, Chen *et al.* reported the CVD growth of large-size MoSe₂ crystals on molten glass at the growth temperature of 1050 °C. This work provides a significant synthesis recipe for the large domain synthesis of TMDs, which is an essential reference for the synthesis of 2D materials, as well as for our work.

In their experimental setup, a piece of SiO₂/Si with a piece of over-positioned Mo foil was used to hold the glass substrate. A small amount of MoO₃ powder was placed below the growth substrate (right on the SiO₂/Si plate). Selenium was placed upstream as the source for the selenization of MoO₃.

The CVD growth was carried out at the temperature of 1050 °C in an atmosphere of mixed Ar/H₂ at an ambient pressure condition.

However, our experimental design is much different from this reference. We suspended the Mo foil on top of glass substrate and introduced a small amount of oxygen to oxidize the Mo foil, leading to the formation of chemically active MoO_{3-x}, finally realized the uniform evaporation and supplement of the Mo sources. Moreover, the growth temperature is set at relative low temperature of ~720 °C. The metal precursor feeding method is overall unique, and rather homogenous on the substrate surface. This can be a typical face-to-face metal precursor supply route, i.e., the Mo-based precursor is released by the mildly oxidized Mo foil that is placed above the substrate in a parallel geometry.

A detailed comparison of the experimental details is listed below in **Table R2** for your check.

Growth conditions	J. Am. Chem. Soc. 139, 1073–1076 (2017)	Adv. Mater. DOI: 10.1002/adma.201704674	Our work
Experimental setup			Precursors	MoO ₃ and Se powders	Mo foils and S powder	Mo foils and S powder
Substrate	Molten soda-lime glass	Molten soda-lime glass	Solid soda-lime glass
Growth temperature	1050° C	1050° C	720° C
Carrier gas	20 sccm Ar and 3 sccm H ₂	20 sccm Ar	50 sccm Ar and 6 sccm O ₂
Pressure	Ambient pressure	Ambient pressure	Low pressure

Table R2 | Comparison of the synthesis methods and growth conditions of Chen *et al.*' work with our work.

Very recently, a newly published work from Chen *et al.* (*Adv. Mater.* doi:10.1002/adma.201704674 (2017)) adopted a similar face-to-face metal precursor supply route by placing a curved Mo foil (Mo source) above the molten glass substrate. At the growth temperature of 1050 °C, the ionic compounds in the glass corroded molybdenum metal by solid-state displacement reactions, which helped with the volatilization of Mo in the pure Ar atmosphere. Single-crystalline MoS₂ flakes with trigonal symmetric patterns were grown homoepitaxially on MoS₂ monolayer and formed a 3R-stacked MoS₂ bilayer.

As compared with the published references, the modifications of our work are as follows (as also listed above as a format, **Table R2**):

- 1) the growth temperature in our system is below the melting point of the soda-lime glass in order to preserve the original surface morphology of the glass substrate;
- 2) the O₂ carrier is introduced to decrease the release temperature of Mo precursor, with the formation of chemically active precursor of MoO_{3-x};
- 3) a large-scale uniformity can be achieved easily with the modified growth method, and larger samples are obtained with the size up to 6-inch; it can be further improved;

4) by exploiting the hydrophilicity feature of the glass substrate, we have developed an etching-free method to transfer our large-area monolayer MoS₂ films onto arbitrary substrates.

According to the reviewer's suggestion, we have cited a new **Ref. 23** and added some discussion in **Page 2** and **Page 3** as follows.

“...Later on, Chen et al. reported the fast-growth of millimetre-size monolayer MoSe₂ crystals on molten glass²². A piece of Mo foil placed on SiO₂/Si was used to hold the glass substrate. Very recently, the same group achieved the homoepitaxial growth of MoS₂ patterns on monolayer MoS₂ at the growth temperature of 1050 °C²³. In the growth process, a piece of curved Mo foil was placed above the molten glass substrate serving as the Mo source. The ionic compounds in glass corroded the Mo foil thus helped with the volatilization of Mo, and a face-to-face metal precursor feeding pathway was resulted....”

“...In contrast, the growth temperature in our system is set to 720 °C in order to preserve the original surface morphology of the glass substrate, thus providing higher potential for the batch production and the direct application of the hybrid material.”

“...The O₂ carrier was introduced to oxidize the Mo foil for directly releasing chemically active MoO_{3-x} (x = 2, 3). This intermediate precursor presents relatively low sublimation temperature of ~500 °C, right below the melting point of the soda-lime glass...”

Reviewers' Comments:

Reviewer #1:

Remarks to the Author:

The authors addressed all the comments properly. I believe the paper can now be published in its current form.

Reviewer #3:

Remarks to the Author:

The authors have addressed all questions well and the paper can be accepted now.

REVIEWERS' COMMENTS:

Reviewer #1 (Remarks to the Author):

The authors addressed all the comments properly. I believe the paper can now be published in its current form.

Our response:

We are very thankful for the reviewer's agreement for publication our manuscript in *Nature Communications*.

Reviewer #3 (Remarks to the Author):

The authors have addressed all questions well and the paper can be accepted now.

Our response:

We are very thankful for the reviewer's agreement for publication our manuscript in *Nature Communications*.